# State and Disturbance Estimation of Autonomous Surface Vehicles based on Nonlinear Cascade Extended State Observers

1st Shijian Jiao
School of Marine Electrical Engineering
Dalian Maritime University
Dalian, China
sjjiao@dlmu.edu.cn

2nd Lu Liu
School of Marine Electrical Engineering
Dalian Maritime University
Dalian, China
luliu@dlmu.edu.cn

3rd Yongqi Yu
School of Marine Electrical Engineering
Dalian Maritime University
Dalian, China
yuy@dlmu.edu.cn

4th Anqing Wang
School of Marine Electrical Engineering
Dalian Maritime University
Dalian, China
anqingwang@dlmu.edu.cn

5th Dan Wang
School of Marine Electrical Engineering
Dalian Maritime University
Dalian, China
dwang@dlmu.edu.cn

6th Zhouhua Peng
School of Marine Electrical Engineering
Dalian Maritime University
Dalian, China
zhpeng@dlmu.edu.cn

*Abstract*—This paper is concerned with the state and disturbance estimation of autonomous surface vehicles (ASVs) in the presence of measurement noises. The velocity state information is unmeasured and the lumped disturbances consisting of internal model uncertainties and external environmental disturbance are unknown. Specifically, in the presence of measurement noises, a nonlinear cascade extended state observer (CESO) based on position-orientation measurement is presented to estimate unmeasured velocity and unknown lumped disturbances. The stability of the overall cascade system is validated through input-to-state stability theory and cascade theory. Simulation results are shown to confirm the effectiveness of the proposed observer scheme.

*Index Terms*—Cascade extended state observer, state estimation, disturbance estimation, measurement noises, autonomous surface vehicles.

## I. INTRODUCTION

Autonomous surface vehicles (ASVs) encounter numerous challenges in their dynamic behavior, primarily due to uncertainties in internal model parameters, unknown hydrodynamic effects, and external disturbance such as wind, waves, and ocean currents [1]–[8]. These factors collectively present substantial obstacles to the motion control design of ASVs. The effectiveness of motion control is heavily dependent on the real-time identification and suppression of lumped disturbances. Consequently, a range of innovative solutions are developed to ensure system stability in environments characterized by coexisting uncertainty and external disturbances [9]–[11].

In [9], an adaptive neural control method based on an innovative event-triggered strategy is proposed to tackle the adaptive neural control problem with dynamic disturbance. In [10], a sliding mode control method based on the extended state observer (ESO) is presented to address the attitude control problem of the quadrotor system under external disturbance. In [11], a nonlinear disturbance observer with auxiliary variable is proposed to guarantee that the error in disturbance estimation remains within a confined and compact range. However, the disturbance estimation methods mentioned in [9]–[11] all require velocity information as a reference.

Due to the simplicity and effectiveness of ESO, the utilization of ESOs has obtained considerable attention in disturbance estimation [12]–[14]. In [12], a finite-time convergent ESO is proposed to estimate the unknown velocities and disturbances. In [13], an event-triggered ESO is designed to prevent redundant communications and cut down communication cost between sensors and observers. In [14], a data-driven adaptive ESO is introduced to estimate unknown input gains absent any pre-existing data of model parameters. Typically, increasing the gain of observers can make it converge more quickly and lessen the impact of disturbance on the steady-state estimation errors. Nevertheless, setting observer gain too high can make the system more sensitive to measurement noises, which in turn increases the estimation error and affects overall noise resistance of the system [15]. However, the methods proposed in [12]–[14] do not consider the effect of measurement noises on ESOs. Several studies tackling the concern are detailed in [16], [17]. In [16], a switched gain observer is designed to estimate the position and velocity in the presence of measurement noises. In [17], a high-gain observer technique using a time-varying gain is designed to reduce the negative impact of measurement noises. However, the observer methods proposed in [16], [17] both require real-time control of the observer gain.

Inspired by the previous discussions, this paper focuses on the challenge of state and disturbance estimation of ASVs in the presence of measurement noises. Specifically, a nonlin-

ear cascade extended state observer (CESO) method based on position-orientation measurement is designed to estimate the velocity state and lumped disturbances for ASVs. Then, stability analysis of the overall cascade system is given. Last, the simulation comparison results of the estimation performance between CESO and classical ESO are presented in the presence of measurement noises.

Compared to previous research efforts in [9]–[14], [16], [17], the key contributions of the paper can be outlined below.

1) In contrast to the disturbance estimation methods discussed in [9]–[11] which rely on velocity measurement, the proposed nonlinear CESO method can recover velocity information and estimate disturbances based on position-heading measurement.

2) In contrast to observer methods in [12]–[14] where the contradiction of balancing the estimation accuracy of high-gain ESOs with noises sensitivity is not considered, the proposed nonlinear CESO can overcome the limitation of measurement noises on classical ESOs.

3) In contrast to observer methods in [16], [17] where the observer gains are changing, the proposed nonlinear CESO method can suppress the measurement noises based on the fixed gain.

The rest of this paper is structured as follows: Section II gives the problem formulation. Section III introduces the observer design. Section IV presents the stability analysis. Section V shows the simulation results. Section VI concludes this paper.

## II. Problem Formulation

Generally, the motion of ASVs on a horizontal plane is characterized using both a earth-fixed coordinate system and a reference frame that is fixed to the body of the vehicle, as shown in Fig. 1. The dynamic of the ASV is defined by

$$\begin{cases} \dot{\eta} = R(\psi)\nu, \\ M\dot{\nu} = -C(\nu)\nu - D(\nu)\nu + g(\nu) + \tau_w + \tau, \end{cases} \quad (1)$$

where

$$R(\psi) = \begin{bmatrix} \cos\psi & -\sin\psi & 0 \\ \sin\psi & \cos\psi & 0 \\ 0 & 0 & 1 \end{bmatrix}, M = \begin{bmatrix} m_u & 0 & 0 \\ 0 & m_v & 0 \\ 0 & 0 & m_r \end{bmatrix};$$

$\eta = [x, y, \psi]^{\mathrm{T}}$ denotes the position and the yaw angle vector; $\nu = [u, v, r]^{\mathrm{T}}$ denotes the surge velocity, sway velocity, and yaw rate; $M$ is the inertial matrix; $C(\nu)$ is the coriolis and centrifugal matrix; $D(\nu)$ is the damping matrix; $g(\nu)$ denotes the unmodelled hydrodynamics; $\tau_w = [\tau_{wu}, \tau_{wv}, \tau_{wr}]^{\mathrm{T}}$ denotes the environmental force; $\tau = [\tau_u, \tau_v, \tau_r]^{\mathrm{T}}$ is the control input.

Define the lumped disturbances $\sigma = [\sigma_u, \sigma_v, \sigma_r]^{\mathrm{T}} = M^{-1}(-C(\nu)\nu - D(\nu)\nu + g(\nu) + \tau_w)$, the ASV dynamic model in (1) can be redefined as

$$\begin{cases} \dot{\eta} = R(\psi)\nu, \\ \dot{\nu} = \sigma + M^{-1}\tau, \end{cases} \quad (2)$$

Although the classical ESO can be used to estimate the unknown lumped disturbances $\sigma$, there may be noise disturbance

in measuring the position-orientation measurement. This paper is focused on the design and implementation of an observer that, in the presence of measurement noises, utilizing position and orientation measurements not only to reconstruct the states of $\nu$ but also to effectively estimate the disturbances $\sigma$.

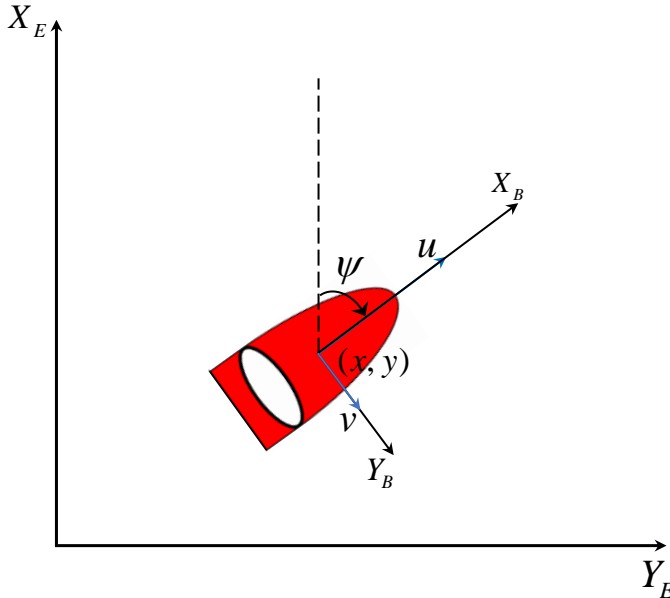

Fig. 1. Geometrical illustration of two reference frames.

## III. Observer Design

In this section, a nonlinear CESO based on position-orientation measurement is designed to estimate the velocity states and lumped disturbances while suppressing measurement noises.

Based on (2), a two-level nonlinear CESO is designed to achieve real-time estimation of $\nu$ and $\sigma$, detailed as follows:

$$\prod_1: \begin{cases} \dot{\hat{\eta}}_1 = -3\omega_1(\hat{\eta}_1 - y_\eta) + R(\psi)\hat{\nu}_1, \\ \dot{\hat{\nu}}_1 = -3\omega_1^2 R^{\mathrm{T}}(\psi)(\hat{\eta}_1 - y_\eta) + \hat{\sigma}_1 + M^{-1}\tau, \\ \dot{\hat{\sigma}}_1 = -\omega_1^3 R^{\mathrm{T}}(\psi)(\hat{\eta}_1 - y_\eta), \end{cases} \quad (3)$$

$$\prod_2: \begin{cases} \dot{\hat{\eta}} = -3\omega_2(\hat{\eta} - \hat{\eta}_1) + R(\psi)\hat{\nu}, \\ \dot{\hat{\nu}} = -3\omega_2^2 R^{\mathrm{T}}(\psi)(\hat{\eta} - \hat{\eta}_1) + \hat{\sigma} + M^{-1}\tau, \\ \dot{\hat{\sigma}}_2 = -\omega_2^3 R^{\mathrm{T}}(\psi)(\hat{\eta} - \hat{\eta}_1), \end{cases} \quad (4)$$

where $y_\eta = \eta + \nu_\eta$ with $\nu_\eta$ being the measurement white noises; $\hat{\eta}_1$, $\hat{\nu}_1$, $\hat{\sigma}_1$, and $\hat{\sigma}_2$ are intermediate variables; $\hat{\sigma}$ is the estimated value of the extended state variable with $\hat{\sigma} = \hat{\sigma}_1 + \hat{\sigma}_2$; $\omega_1$ and $\omega_2$ are the positive observer gains.

Let

$$\begin{cases} \tilde{\eta}_1 = \hat{\eta}_1 - \eta, s_1 = \tilde{\eta}_1(\omega_1 t)\omega_1^2, \\ \tilde{\nu}_1 = \hat{\nu}_1 - \nu, s_2 = \tilde{\nu}_1(\omega_1 t)\omega_1, \\ \tilde{\sigma}_1 = \hat{\sigma}_1 - \sigma, s_3 = \tilde{\sigma}_1(\omega_1 t), \end{cases} \quad (5)$$

it renders that

$$\begin{cases} \dot{s}_1 = -3s_1 + R(\psi)s_2 + 3\omega_1^2\nu_\eta, \\ \dot{s}_2 = -3R^{\mathrm{T}}(\psi)s_1 + s_3 + 3\omega_1^2 R^{\mathrm{T}}(\psi)\nu_\eta, \\ \dot{s}_3 = -R^{\mathrm{T}}(\psi)s_1 - \frac{\dot{\sigma}_1}{\omega_1} + \omega_1^2 R^{\mathrm{T}}(\psi)\nu_\eta. \end{cases} \quad (6)$$

Define $S_1 = [s_1^\mathrm{T}, s_2^\mathrm{T}, s_3^\mathrm{T}]^\mathrm{T} \in \Re^9$, the error dynamics (6) can be denoted by

$$\dot{S}_1 = AS_1 - B\frac{\dot{\sigma}_1}{\omega_1} + C\omega_1^2\nu_\eta, \qquad (7)$$

where

$$A = \begin{bmatrix} -3I_3 & R(\psi) & 0_3 \\ -3R^\mathrm{T}(\psi) & 0_3 & I_3 \\ -R^\mathrm{T}(\psi) & 0_3 & 0_3 \end{bmatrix},$$

$$B = \begin{bmatrix} 0_3 \\ 0_3 \\ I_3 \end{bmatrix}, C = \begin{bmatrix} 3I_3 \\ 3R^\mathrm{T}(\psi) \\ R^\mathrm{T}(\psi) \end{bmatrix}.$$

Let

$$\begin{cases} \tilde{\eta} = \hat{\eta} - \eta, s_4 = \tilde{\eta}(\omega_2 t)\omega_2^2, \\ \tilde{\nu} = \hat{\nu} - \nu, s_5 = \tilde{\nu}(\omega_2 t)\omega_2, \\ \tilde{\sigma}_2 = \hat{\sigma}_2 - \sigma, s_6 = \tilde{\sigma}_2(\omega_2 t), \end{cases} \qquad (8)$$

it renders that

$$\begin{cases} \dot{s}_4 = -3s_4 + R(\psi)s_5 + 3\omega_2^2\tilde{\eta}_1 \\ \dot{s}_5 = -3R^\mathrm{T}(\psi)s_4 + s_6 + 3\omega_2^2 R^\mathrm{T}(\psi)\tilde{\eta}_1 \\ \dot{s}_6 = -R^\mathrm{T}(\psi)s_4 - \frac{\dot{\sigma}_2}{\omega_2} + \omega_2^2 R^\mathrm{T}(\psi)\tilde{\eta}_1. \end{cases} \qquad (9)$$

Define $S_2 = [s_4^\mathrm{T}, s_5^\mathrm{T}, s_6^\mathrm{T}]^\mathrm{T} \in \Re^9$, the error dynamics (9) can be denoted by

$$\dot{S}_2 = AS_2 - B\frac{\dot{\sigma}_2}{\omega_2} + C\omega_2^2\tilde{\eta}_1, \qquad (10)$$

A transformation characterized by block-diagonal properties is hereby presented:

$$\begin{cases} E_1 = QS_1 \\ E_2 = QS_2, \end{cases} \qquad (11)$$

with $Q = \mathrm{diag}\left\{R^\mathrm{T}(\psi), I_3, I_3\right\}$.

By combining (7), (10), and (11), it can be obtained that

$$\dot{E}_1 = A_0 E_1 + kZ_\mathrm{T} E_1 - B\frac{\dot{\sigma}_1}{\omega_1} + C_0\omega_1^2\nu_\eta, \qquad (12)$$

$$\dot{E}_2 = A_0 E_2 + kZ_\mathrm{T} E_2 - B\frac{\dot{\sigma}_2}{\omega_2} + C_0\omega_2^2\tilde{\eta}_1, \qquad (13)$$

with $Z_\mathrm{T} = \mathrm{diag}\left\{Z^\mathrm{T}, 0_3, 0_3\right\}$, and

$$A = \begin{bmatrix} -3I_3 & I_3 & 0_3 \\ -3I_3 & 0_3 & I_3 \\ -I_3 & 0_3 & 0_3 \end{bmatrix},$$

$$Z = \begin{bmatrix} 0 & -1 & 0 \\ 1 & 0 & 0 \\ 0 & 0 & 0 \end{bmatrix}, C_0 = \begin{bmatrix} 3I_3 \\ 3I_3 \\ I_3 \end{bmatrix}.$$

The formula (7) can be viewed as a system defined by the state vector $S_1$ and the inputs $\dot{\sigma}_1$, $\nu_\eta$; the formula (10) can be viewed as a system defined by the state vector $S_2$ and the inputs $\dot{\sigma}_2$, $\tilde{\eta}_1(S_1)$. Thus, $(S_1, S_2)$ can be regarded as a cascade system.

*Remark 1*: From (3) and (4), only $\prod_1$ is influenced by measurement noises. In contrast, $\prod_2$ relies solely on the estimates from $\prod_1$ as its benchmark. This two-level approach effectively reduces the noise impacting of the final estimates

in $\prod_2$. Additionally, the gain of the observer for $\prod_1$ is intentionally set lower than that of $\prod_2$, the gain of the observer for $\prod_2$ corresponds to the classical ESO, which helps to minimize the noise affecting of $\prod_1$.

## IV. STABILITY ANALYSIS

In Section 3, a nonlinear CESO algorithm is proposed. This section analyzes the stability of the overall cascade system.

*Assumption 1*: $\|\dot{\sigma}_1\| \leq \sigma_1^*$, $\|\dot{\sigma}_2\| \leq \sigma_2^*$ with $\sigma_1^*, \sigma_2^* \in \mathbb{R}^+$.

A theorem is presented to summarize the result discussed in Section 3 as follows: *Theorem 1*: The cascade system $(S_1, S_2)$ with the inputs $\dot{\sigma}_1, \nu_\eta, \dot{\sigma}_2$ consisting of subsystem (7) and subsystem (10) is input-to-state stable (ISS).

*Proof:* Construct the following Lyapunov function

$$V_1 = \frac{1}{2}E_1^\mathrm{T} P_1 E_1 \qquad (14)$$

where $P_1$ is a positive definite matrix, it satisfies the following simultaneous Lyapunov inequalities [18]

$$\begin{cases} A_0^\mathrm{T} P + P A_0 + \beta I \leq k^* \left(Z_\mathrm{T}^\mathrm{T} P + P Z_\mathrm{T}\right), \\ A_0^\mathrm{T} P + P A_0 + \beta I \leq -k^* \left(Z_\mathrm{T}^\mathrm{T} P + P Z_\mathrm{T}\right), \end{cases} \qquad (15)$$

where $\beta \in \mathbb{R}^+$ and $k^* \in \mathbb{R}^+$ complying with $|k| \leq k^*$.

Deriving the derivative of $V_1$ over time and employing (12), one has:

$$\dot{V}_1 = \frac{1}{2}E_1^\mathrm{T}(P_1 A_0 + A_0^\mathrm{T} P_1 + k(P_1 Z_\mathrm{T} + Z_\mathrm{T}^\mathrm{T} P_1))E_1 \\ + E_1^\mathrm{T} P_1 B(-\frac{\dot{\sigma}_1}{\omega_1}) + E_1^\mathrm{T} P_1 C_0(\omega_1^2\nu_\eta). \qquad (16)$$

Obviously, band-limited white nosie $\nu_\eta$ is bounded. Substituting (15) into (16), one has:

$$\dot{V}_1 \leq -\frac{\beta}{2}\|E_1\|^2 + \frac{\|E_1\|\|P_1 B\|\|\dot{\sigma}_1\|}{\omega_1} + \omega_1^2\|E_1\|\|P_1 C_0\|\|\nu_\eta\|. \qquad (17)$$

Since

$$\|E_1\| \geq \frac{2\|P_1 B\|\|\dot{\sigma}_1\|}{\omega_1\beta\rho_1} + \frac{2\omega_1^2}{\beta\rho_1}\|P_1 C_0\|\|\nu_\eta\|, \qquad (18)$$

it results in

$$\dot{V}_1 \leq -\frac{\beta}{2}\left(1 - \rho_1\right)\|E_1\|^2 \qquad (19)$$

with $0 < \rho_1 < 1$. It implies that the system (12) is ISS, and there exists a $\mathcal{KL}$ function $\alpha$ and a $\mathcal{K}$ function $\Upsilon$ complying with

$$\|E_1(t)\| \leq \max\{\alpha(\|E_1(0)\|, t), \Upsilon(|\dot{\sigma}_1|) + \Upsilon(|\nu_\eta|)\} \qquad (20)$$

with

$$\Upsilon(s) = \sqrt{\frac{\lambda_{\max}(P_1)}{\lambda_{\min}(P_1)}}\left(\frac{2\|P_1 B\|s}{\omega_1\beta\rho_1} + \frac{2\omega_1^2\|P_1 C_0\|s}{\beta\rho_1}\right). \qquad (21)$$

Note that $\|E_1\| = \|S_1\|$, it renders that

$$\|S_1(t)\| \leq \max\{\alpha(\|S_1(0)\|, t), \Upsilon(|\dot{\sigma}_1|) + \Upsilon(|\nu_\eta|)\}, \qquad (22)$$

thus, the subsystem (7) is ISS.

Construct the following Lyapunov function

$$V_2 = \frac{1}{2} E_2^{\mathrm{T}} P_2 E_2 \qquad (23)$$

where $P_2$ is a positive definite matrix, it satisfy (15), similar to $P_1$.

Taking the time derivative of $V_2$ and using (13), one has:

$$\dot{V}_2 = \frac{1}{2} E_2^{\mathrm{T}} (P_2 A_0 + A_0^{\mathrm{T}} P_2 + k(P_2 Z_{\mathrm{T}} + Z_{\mathrm{T}}^{\mathrm{T}} P_2)) E_2 \\ + E_2^{\mathrm{T}} P_2 B(-\frac{\dot{\sigma}_2}{\omega_2}) + E_2^{\mathrm{T}} P_2 C_0(\omega_2^2 \tilde{\eta}_1). \qquad (24)$$

Substituting (15) into (24), one has:

$$\dot{V}_2 \leq -\frac{\beta}{2} \|E_2\|^2 + \frac{\|E_2\| \|P_2 B\| \|\dot{\sigma}_2\|}{\omega_2} + \omega_2^2 \|E_2\| \|P_2 C_0\| \|\tilde{\eta}_1\|. \qquad (25)$$

Since

$$\|E_2\| \geq \frac{2\|P_2 B\| \|\dot{\sigma}_2\|}{\omega_2 \beta \rho_2} + \frac{2\omega_2^2}{\beta \rho_2} \|P_2 C_0\| \|\tilde{\eta}_1\|, \qquad (26)$$

it results in

$$\dot{V}_2 \leq -\frac{\beta}{2}(1 - \rho_2) \|E_2\|^2 \qquad (27)$$

with $0 < \rho_2 < 1$. It implies that the system (13) is ISS, and there exists a $\mathcal{KL}$ function $\zeta$ and a $\mathcal{K}$ function $\Gamma$ complying with

$$\|E_2(t)\| \leq \max\{\zeta(\|E_2(0)\|, t), \Gamma(|\dot{\sigma}_2|) + \Gamma(|\tilde{\eta}_1|)\} \qquad (28)$$

with

$$\Gamma(s) = \sqrt{\frac{\lambda_{\max}(P_2)}{\lambda_{\min}(P_2)}} \left( \frac{2\|P_2 B\| s}{\omega_2 \beta \rho_2} + \frac{2\omega_2^2 \|P_2 C_0\| s}{\beta \rho_2} \right). \qquad (29)$$

Note that $\|E_2\| = \|S_2\|$, it renders that

$$\|S_2(t)\| \leq \max\{\zeta(\|S_2(0)\|, t), \Gamma(|\dot{\sigma}_2|) + \Gamma(|\tilde{\eta}_1|)\}, \qquad (30)$$

thus, the subsystem (10) is ISS.

Due to the subsystem (7) and the subsystem (10) being ISS, Lemma 4.6 in [19] shows that the two subsystems are referred to a cascade system $(S_1, S_2)$ concerning the inputs $\dot{\sigma}_1$, $\nu_\eta$, $\dot{\sigma}_2$, and the cascade system is ISS. ∎

## V. SIMULATION RESULTS

In this section, the classical nonlinear ESO and the proposed nonlinear CESO are utilized to verify and compare the effectiveness of velocity state and disturbance estimation for surface vehicles, respectively. The parameters of nonlinear CESO algorithm are referenced in Table I.

Simulation results are depicted in Figs. 2-13. In this paper, two observer gain modes are designed. Figs. 2-7 show the state and disturbance estimation effects of classical nonlinear ESO and nonlinear CESO in the case of $\omega_1 = 10, \omega_2 = 30$. It can be seen that in the presence of measurement noises, the nonlinear CESO has better estimation performance compared to the nonlinear ESO. Figs. 8-13 shows the state and disturbance estimation of classical nonlinear ESO and nonlinear CESO in the case of $\omega_1 = 10, \omega_2 = 40$. It can be seen that the increase

in the gain of observers amplifies the impact of measurement noises on both state and disturbance estimation performance, which reflects the superiority of the proposed nonlinear CESO in suppressing the measurement noises.

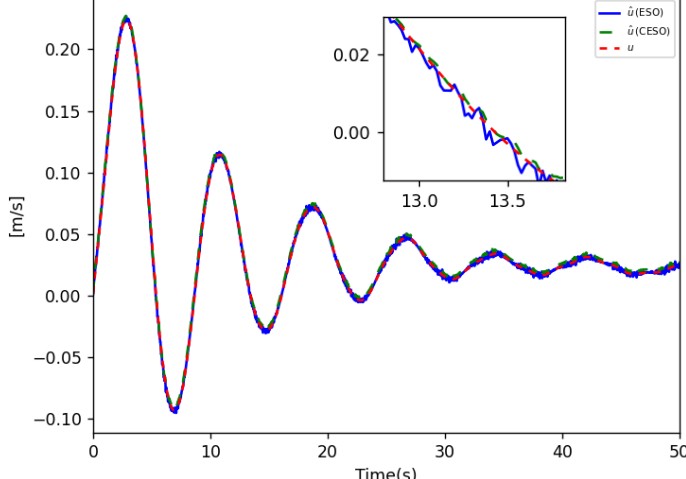

Fig. 2. Surge velocity estimation($\omega_1 = 10, \omega_2 = 30$).

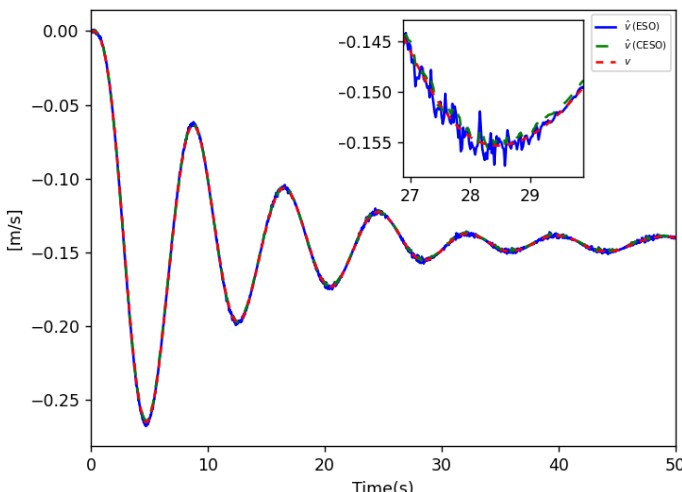

Fig. 3. Sway velocity estimation($\omega_1 = 10, \omega_2 = 30$).

TABLE I
SIMULATION PARAMETERS OF NONLINEAR CESO ALGORITHM

| Parameter | Value |
|-----------|-------|
| $m_u$ | 25.8 |
| $m_v$ | 33.8 |
| $m_r$ | 2.76 |
| $\omega_1$ | 10 |
| $\omega_2$ | 30, 40 |
| $\tau_{wu}$ | $-0.2\cos(t)\cos(1.5t)$ |
| $\tau_{wv}$ | $0.1\sin(0.1t)$ |
| $\tau_{wr}$ | $-0.3\sin(2t)\cos(2.3t)$ |

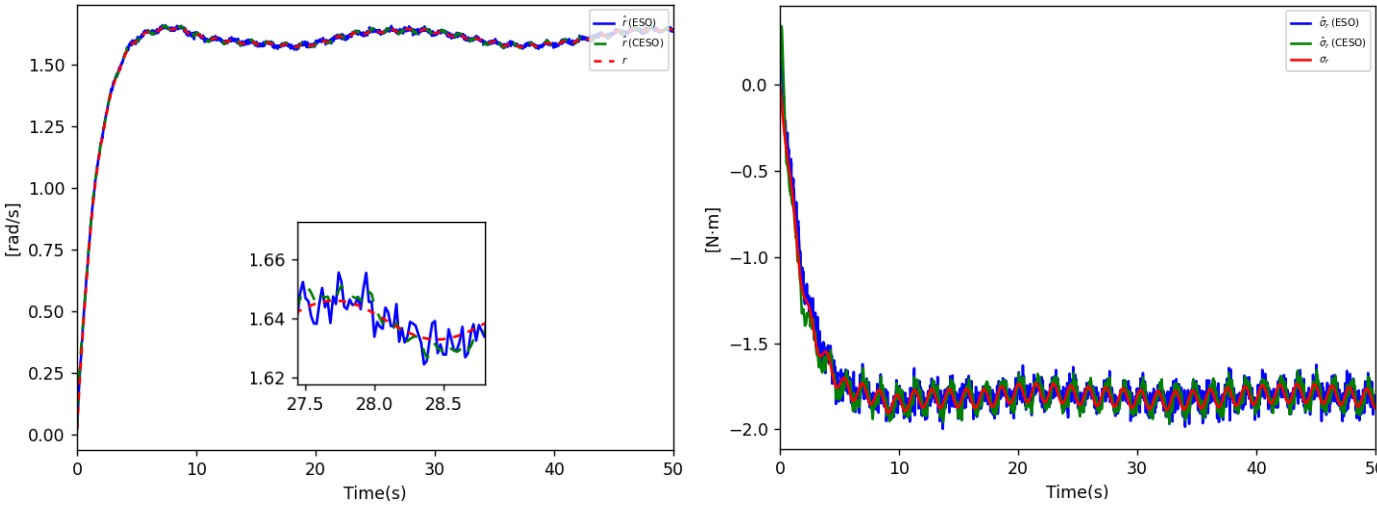

Fig. 4. Yaw rate estimation($\omega_1 = 10$, $\omega_2 = 30$).

Fig. 7. Disturbance estimation in the yaw direction($\omega_1 = 10$, $\omega_2 = 30$).

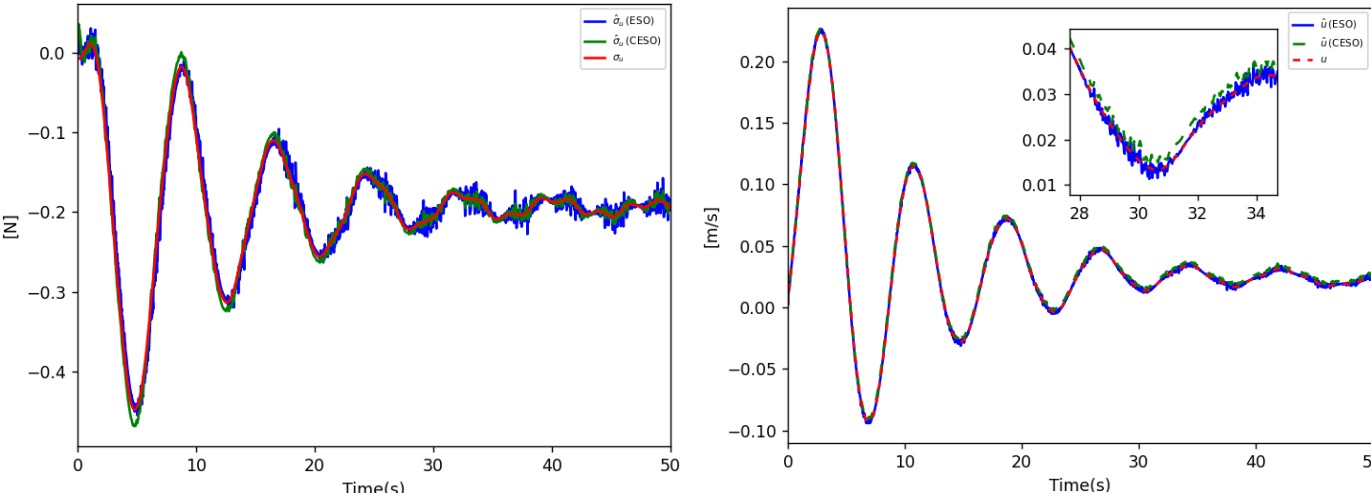

Fig. 5. Disturbance estimation in the surge direction($\omega_1 = 10$, $\omega_2 = 30$).

Fig. 8. Surge velocity estimation($\omega_1 = 10$, $\omega_2 = 40$).

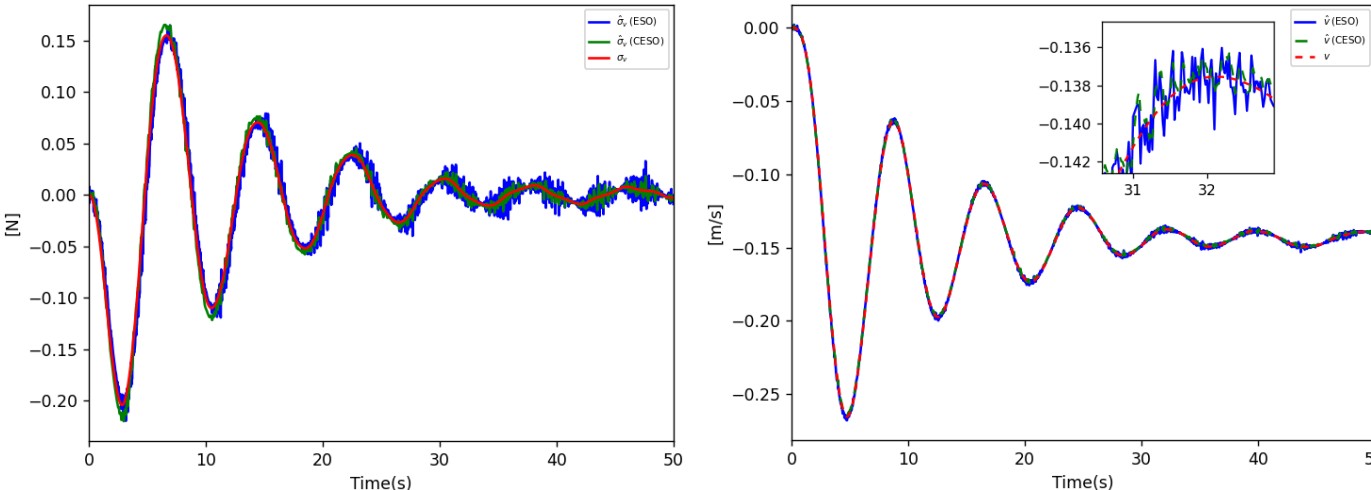

Fig. 6. Disturbance estimation in the sway direction($\omega_1 = 10$, $\omega_2 = 30$).

Fig. 9. Sway velocity estimation($\omega_1 = 10$, $\omega_2 = 40$).

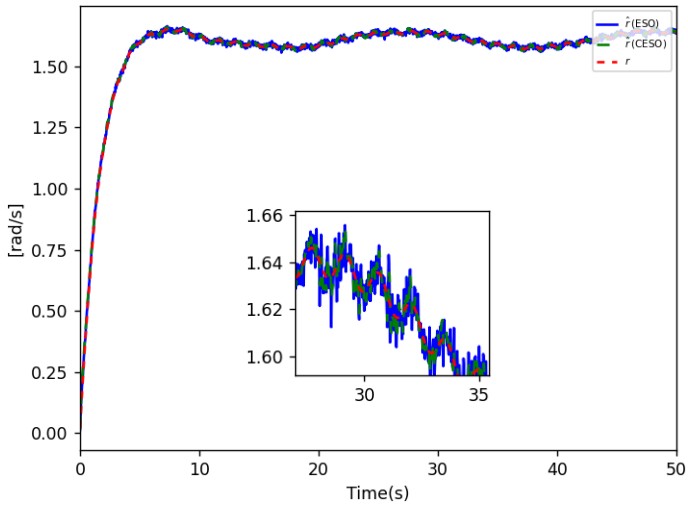

Fig. 10. Yaw rate estimation($\omega_1 = 10$, $\omega_2 = 40$).

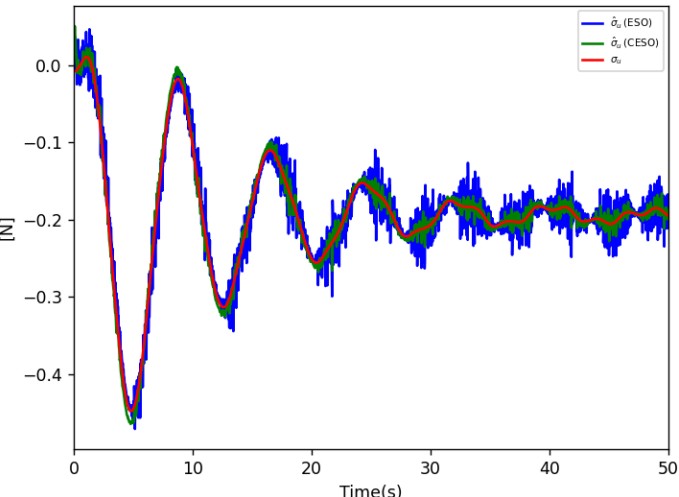

Fig. 11. Disturbance estimation in the surge direction($\omega_1 = 10$, $\omega_2 = 40$).

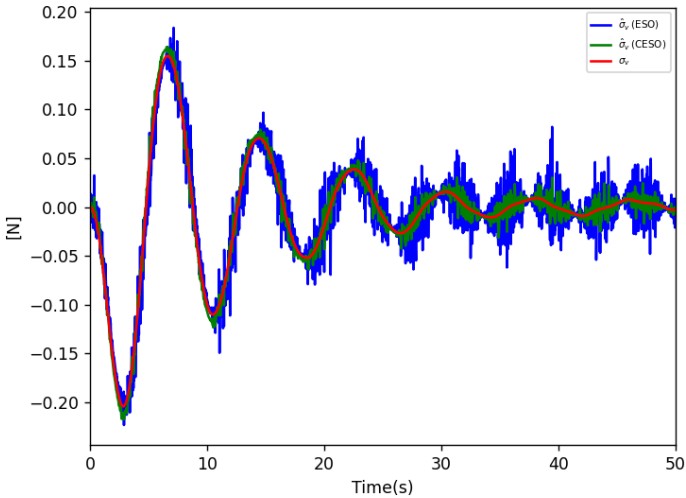

Fig. 12. Disturbance estimation in the sway direction($\omega_1 = 10$, $\omega_2 = 40$).

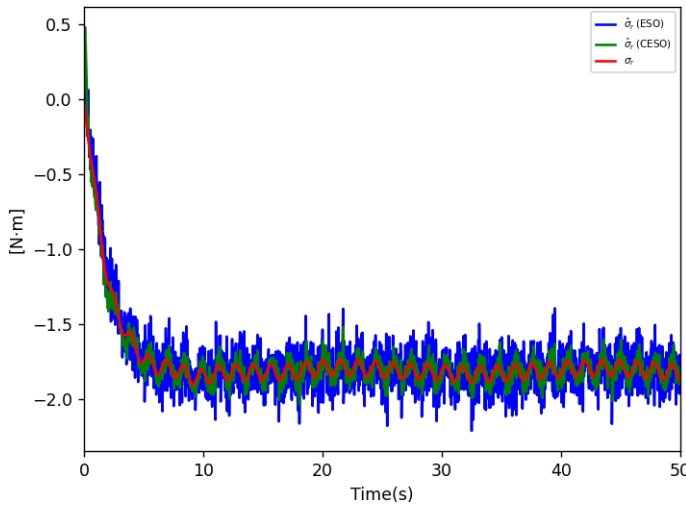

Fig. 13. Disturbance estimation in the yaw direction($\omega_1 = 10$, $\omega_2 = 40$).

## VI. CONCLUSION

In this paper, a state and disturbance estimation method is introduced for ASVs in the presence of measurement noises. A nonlinear CESO method based on position-orientation measurement is proposed to extract the unmeasured velocity and lumped disturbances. The stability of the overall cascade system is confirmed by utilizing the input-to-state stability and cascade theory. Simulation results substantiate the effectiveness of the proposed method in this paper.

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
