# OpenReview forum: "State and Disturbance Estimation of Autonomous Surface Vehicles based on Nonlinear Cascade Extended State Observers"
_IEEE.org/ICIST/2024/Conference — IEEE ICIST 2024 Conference Submission_

### Official Review · Reviewer_Pp4g · 2024-08-26
**This paper is concerned with the state and disturbance estimation of autonomous surface vehicles (ASVs) in the presence of measurement noises. Meanwhile, the simulation results verified the effectiveness of the method. Here are some comments.**

**Rating:** 7
**Confidence:** 3

**Review:**

Question 1:
How does the nonlinear CESO method address the challenges of velocity estimation and disturbance recovery based on position-heading measurements compared to the velocity-based methods discussed in [9]-[11]?
Question 2:
What specific mechanisms in the proposed nonlinear CESO method enable it to mitigate the trade-off between estimation accuracy and noise sensitivity, which is a challenge in high-gain ESOs as highlighted in [12]-[14]?
Question 3:
In what ways does the fixed-gain approach of the proposed nonlinear CESO method improve noise suppression compared to the variable-gain strategies employed in observer methods from [16] and [17]?

---

### Official Review · Reviewer_7kYs · 2024-08-26
**The scheme is effective, the simulation results are reasonable, and it is recommended to publish.**

**Rating:** 7
**Confidence:** 4

**Review:**

The paper presents an effective approach to the state and disturbance estimation problem for autonomous surface vehicles (ASVs) under the influence of measurement noise. The article is well-structured and the simulation results are reliable. The reviewer has the following questions to discuss with the authors:

1. How does the proposed CESO compare with other existing state observers in terms of accuracy and computational efficiency?
2. How sensitive is the proposed observer to different levels of measurement noise?
3. Can the proposed observer be adapted for use in other types of autonomous vehicles, such as aerial or underwater vehicles?

---

### Official Review · Reviewer_ouQY · 2024-08-28
**The paper is logically clear, the simulation results are credible, and it is recommended for publication**

**Rating:** 7
**Confidence:** 3

**Review:**

This paper is concerned with the state and disturbance estimation of autonomous surface vehicles (ASVs) in the presence of measurement noises. This paper is well written. The following comment need to be answered to further improve the quality of this paper.
1.While the proposed nonlinear cascade extended state observer (CESO) effectively estimates states and disturbances in the presence of measurement noises, its performance could still be impacted by model uncertainties. Future work could incorporate adaptive techniques to tune the observer gains based on real-time estimation of model uncertainties. This could enhance the robustness of the observer and further improve the estimation accuracy.
2.The current study focuses on estimating lumped disturbances that are assumed to be bounded but with constant upper bounds. However, in real-world applications, disturbances like wind, waves, and ocean currents can be time-varying. Incorporating adaptive or predictive methods to handle time-varying disturbances could significantly improve the practical applicability of the proposed CESO.
3.The current work focuses on a relatively simple ASV dynamics model with three degrees of freedom (surge, sway, and yaw). For more complex autonomous surface vehicles with additional dynamic degrees of freedom (e.g., heave, roll, and pitch), extending the CESO to higher dimensional systems would be valuable. This would require modifications to the observer structure and stability analysis to ensure robust estimation performance in higher dimensional spaces.

---

### Decision · Program_Chairs · 2024-09-08

Accept (Oral)